# Effect of 8-Hydroxyguanine DNA Glycosylase 1 on the Function of Immune Cells

**DOI:** 10.3390/antiox12061300

**Published:** 2023-06-19

**Authors:** Weiran Zhang, Ranwei Zhong, Xiangping Qu, Yang Xiang, Ming Ji

**Affiliations:** Department of Physiology, School of Basic Medicine, Central South University, Changsha 410078, China; 216511004@csu.edu.cn (W.Z.);

**Keywords:** OGG1, oxidative damage, base excision repair, immune cells

## Abstract

Excess reactive oxygen species (ROS) can cause an imbalance between oxidation and anti-oxidation, leading to the occurrence of oxidative stress in the body. The most common product of ROS-induced base damage is 8-hydroxyguanine (8-oxoG). Failure to promptly remove 8-oxoG often causes mutations during DNA replication. 8-oxoG is cleared from cells by the 8-oxoG DNA glycosylase 1 (OGG1)-mediated oxidative damage base excision repair pathway so as to prevent cells from suffering dysfunction due to oxidative stress. Physiological immune homeostasis and, in particular, immune cell function are vulnerable to oxidative stress. Evidence suggests that inflammation, aging, cancer, and other diseases are related to an imbalance in immune homeostasis caused by oxidative stress. However, the role of the OGG1-mediated oxidative damage repair pathway in the activation and maintenance of immune cell function is unknown. This review summarizes the current understanding of the effect of OGG1 on immune cell function.

## 1. Introduction

Reactive oxygen species (ROS) are common metabolites that can be physiologically cleared. However, oxidative stress results in the excessive production and accumulation of ROS. Biomacromolecules, such as DNA, can be damaged by ROS, resulting in impaired cell function or even apoptosis or necrosis. This ultimately leads to an imbalance in the functional homeostasis of an organism and disease onset.

The DNA damage response and repair (DDR/R) network can identify and repair DNA damage, ensure the correct transmission of genetic material during the stages of DNA replication, repair, recombination, and chromosome segregation, and enhance the body’s ability to cope with oxidative stress damage caused by endogenous or exogenous stimuli [1,2]. Accumulating evidence supports an interaction between DNA damage repair processes and immune cells. On the one hand, excessive DNA oxidative damage leads to dysfunction of the DDR/R network, resulting in the accumulation of single-stranded DNA and double-stranded DNA in the cytoplasm; this triggers the activation of the immune system by inducing the cGAS-STING-IRF3 pathway and producing type I interferon (IFN) [3]. On the other hand, immune cell dysfunction or secretion of large amounts of inflammatory factors can also cause DNA damage and activate the DDR/R network [4]. Disruption of immune cell function is usually accompanied by excess production of ROS and reactive nitrogen species (RNS), abnormal immune responses to autoantigens, and oxidative stress [5]. This suggests that there is a close link between oxidative damage and immune cell dysfunction.

DNA oxidative damage can be repaired by the base excision repair (BER) pathway, which is highly conserved in both eukaryotes and prokaryotes [6]. 8-hydroxyguanine (8-oxoG) is the most common product of this pathway and a biomarker of DNA oxidative damage [7,8,9]. In vivo, 8-oxoG is specifically recognized and eliminated by 8-oxoG DNA glycosylase 1 (OGG1). OGG1 can repair DNA oxidative damage by clearing 8-oxoG, thereby maintaining the stability of DNA and enzymes that repair gene expression [10]. *Ogg1^−/−^* mice were found to resist endogenous toxicity-induced organ dysfunction, neutrophil infiltration, and oxidative stress, but at the same time, *Ogg1^−/−^* mice fed a high-fat diet exhibited higher fasting plasma insulin levels and increased susceptibility to obesity and diabetes [11,12]. This review summarizes the effects of OGG1 on various immune cell functions and discusses novel research concepts and treatment targets for the pathogenesis of clinical inflammatory diseases.

## 2. The Role of OGG1 in DNA Oxidative Damage Repair

Oxidative stress can cause DNA strand breakage and base modifications, significantly accelerating DNA damage. DNA oxidative damage is responsible for the development of many diseases, including chronic inflammation [13], cancer [14], and premature aging [15]. Therefore, the ability of cells to effectively repair DNA damage is crucial for maintaining functional cell homeostasis. Among the four bases, guanine (G) has the lowest redox potential and is the most easily oxidized base [16]. Guanine is oxidized to form 8-oxoG, which pairs with adenine (A) during base complementary pairing, resulting in a G:C-A:T base reversal mutation after two DNA replications [8] (Figure 1). 8-oxoG accounts for 5% of the total number of known oxidized DNA bases and is a marker of DNA oxidative damage [17].

The DDR system is initiated to coordinate the activation of DNA damage checkpoints and promote the clearance of DNA damage in order to maintain genome integrity and stability. DDR processes include the direct reversal of certain types of damage, such as enzymatic photoreactivation of thymine dimers, and the excision of damaged bases via BER, nucleotide excision repair (NER), and mismatch repair (MMR) [18]. Among these pathways, BER is the main pathway for oxidative damage repair. In the BER pathway, based on the type of DNA damage, specific DNA glycosylases catalyze the hydrolysis of the N-glycosidic bond of damaged deoxyribonucleosides to form a depurine/depyrimidine (AP) site on the DNA strand. The AP endonuclease then cuts a short segment of the DNA, including the AP site, leading to the formation of a newly repaired fragment via DNA polymerase and ligase activity [19].

In mammalian cells, 8-oxoG is normally cleared through the OGG1-mediated BER pathway. OGG1 is a functional analog of the *Escherichia coli* protein MutM/Fpg, which removes 8-oxoG and its ring-opening product FapyG from the DNA duplex [20]. When guanine on the DNA strand is damaged by ROS, OGG1 can specifically recognize and excise 8-oxoG, a product of DNA oxidative damage. During this process, OGG1 utilizes its AP lyase activity to break the 3′ phosphodiester bond of the nucleotide at the cleavage site to form a base-free gap, with 3′-dRP forming upstream and 5′-P forming downstream of the gap. Subsequently, OGG1 recruits AP nucleic acid endonuclease 1 (APE1) to the location, which clears 3′-dRP upstream of the gap to form 3′-OH. This allows for the correct base to be introduced by DNA polymerases, and the gap is joined by DNA ligase to reform a complete DNA strand [21].

Given that the occurrence of inflammation is closely related to ROS and the oxidative DNA damage product 8-oxoG [22], what role does the OGG1-mediated base repair pathway play in various inflammatory responses? Analysis of genome-wide data using RNA-seq techniques has revealed that, in airway epithelial cells, OGG1 recognizes 8-oxoG excised from DNA and forms a complex with the excised 8-oxoG. Then, the complex acts as a guanine nucleotide exchange factor, activating small GTPases. This initiates a signaling cascade reaction to translocate transcription factors, thereby promoting the transcription of genes associated with airway hyperreactivity, mast cell degranulation, and bronchoconstriction [23]. OGG1 can also induce 8-oxoG-triggered activation of dendritic cells (DCs) by initiating a small molecule of GTPase [24]. In addition, in mouse asthma models induced by ovalbumin (OVA), TH5487, a small molecule inhibitor of OGG1, was found to significantly inhibit plasma levels of Immunoglobulin E(IgE) and the recruitment of eosinophils and other immune cells to the lungs, while also downregulating the expression of tumor necrosis factor receptor superfamily member 4, arginase 1, Chemokine (C-C motif) ligand 12 and Chemokine (C-C motif) ligand 11, and decreasing levels of activated nuclear factor kappa-B(NF-κB) [25,26,27]. Another study found that *Ogg1^−/−^* mice had a large increase in oxidized mitochondrial DNA (ox-mtDNA) due to *Ogg1* deficiency, which caused the activation of NLRP3 inflammatory vesicles and promoted IL-1β secretion, accelerating the development of atherosclerosis [28]. In contrast, in bone marrow-derived macrophages (BMDMs), mitochondrial OGG1 was found to inhibit NLRP3 activity and the activation of the cyclic GMP-AMP synthase stimulator of interferon genes (cGAS-STING) pathway through the repair of Ox-mtDNA, thereby attenuating lipopolysaccharide (LPS)-induced lung inflammation [29]. Together, the above findings suggest that OGG1 not only mediates the repair of oxidative DNA damage to maintain the homeostasis of cellular functions, but also acts as an important cell signaling molecule and is involved in the regulation of inflammation-related gene expression and the activation of related signaling pathways.

In recent years, studies have revealed that OGG1 not only participates in the traditional BER pathway, but its small molecule activator TH10785 also interacts with OGG1 to activate the β,δ-cleavage enzyme function of OGG1; in this process, the clearance of AP sites by OGG1 is no longer dependent on APE1, but on the repair of polynucleotide kinase phosphatase (PNKP1) activity. TH10785 has also been found to significantly increase intracellular OGG1 recruitment and repair oxidative DNA damage [30].

In addition, OGG1 can act as a signal transduction molecule to regulate gene transcription and, thus, protect against oxidative stress. In one study, researchers examined 3-methylcholanthrene (3-MCA)-induced lung cancer in a rat model and human lung cancer tissue samples and found that TET1 methylation levels were significantly upregulated while DNA hydroxymethylation levels in the promoter regions of XRCC1, OGG1, and APEX1, which are key genes of the BER pathway, were significantly reduced. Moreover, methylation levels gradually increased. Blocking OGG1 significantly attenuated the effect of TET1 [31]. This is due to the ability of OGG1 to target and bind to 8-oxoG and then recruit the dioxygenase TET1 to oxidize 5-methylcytosine (5-mC) adjacent to 8-oxoG to form 5-hydroxymethylcytosine (5-hmC), thereby initiating DNA demethylation [32] (Figure 1). These studies have provided insight into the mechanism of action of OGG1 and have highlighted the need for a more comprehensive understanding and assessment of the role of OGG1 in diseases associated with oxidative DNA damage.

## 3. The Role of OGG1 in Immune Cell Activation

The maintenance of immune homeostasis depends on the normal function of immune cells [33]. Evidence suggests that oxidative stress is closely related to immune cell function. For example, some immune cells in the innate immune system can produce ROS to kill pathogens. However, long-term inflammatory processes can lead to an imbalance in the redox homeostasis of immune cells, resulting in excessive ROS production and oxidative damage [34]. In respiratory syncytial virus (RSV)-induced lung and respiratory tract inflammation in mice, the presence of *Ogg1* exacerbates lung inflammation and results in histopathological changes in mice, whereas intervention with TH5487 reduces the RSV-induced expression of associated chemokines and cytokines and significantly reduces the associated clinical signs [35]. These results suggest that OGG1 plays a very important role in the immune response.

### 3.1. OGG1 and Macrophages

Innate immune responses are mainly activated by phagocytosis, either by non-professional immune cells (e.g., epithelial cells, endothelial cells, and fibroblasts) or by professional antigen-presenting cells (APCs) (e.g., neutrophils, macrophages, dendritic cells, and B cells) [36]. A previous study reported that oxidative DNA damage is present in vascular smooth muscle cells (VSMCs) and macrophages in atherosclerotic plaques, accompanied by the in situ accumulation of 8-oxoG in macrophages during disease progression [37]. This suggests that macrophage function may be affected by exposure to oxidative stress.

Kumar et al. found that high-glucose (HG) treatment increased the accumulation of ROS and RNS in macrophages, inducing apoptosis. In addition, the levels of superoxide dismutase and glutathione, which are related to free radical scavenging, were reduced in macrophages, suggesting that this oxidative damage was induced by HG. Western blot results also showed that OGG1 expression reduced and the phosphorylation levels of Akt and tuberin rapidly increased in macrophages after HG induction via the ERK1/2 MAPK pathway. Subsequently, the treatment of HG-induced macrophages using exogenous antioxidants resulted in the upregulation of OGG1 expression, a reduction in 8-oxoG accumulation, and the downregulation of the inflammatory cytokines TNF-α, IL-1α, and CXCL-10 and the chemokine CCL-8 [38,39]. However, the mechanisms by which OGG1 affects macrophage activation and antigen presentation are not yet clear and warrant further exploration.

### 3.2. OGG1 and Dendritic Cells

DCs are intermediate players in antigen presentation and are extremely important for both innate and acquired immunity. DCs undergo functional changes in response to oxidative stress. Pazmandi et al. examined the effects of normal mitochondrial DNA (mtDNA) and oxidatively damaged mtDNA on the phenotype and function of plasmacytoid DCs (pDCs), respectively. This study found that normal mtDNA led to an upregulation in the expression of CD86, CD83, and the major antigen-presenting molecule human leukocyte antigen (HLA)-DQ, as well as significant increases in the secretion of TNF-α and IL-8 by pDCs. These effects were more obvious in pDCs with oxidatively damaged mtDNA [40]. These results were further validated in vivo using appropriate animal models. The above study was the first to demonstrate that increased levels of 8-oxoG in mtDNA outside of the nucleus, caused by oxidative stress, enhance the stimulatory capacity of mtDNA on pDCs, leading to increased pDC activation and upregulated expression of associated cytokines. Notably, the effect of mtDNA on human pDCs was abrogated by the administration of Toll-like receptor 9 (TLR9) antagonists [40,41].

To explore the specific mechanism of DC activation by ROS, exogenous 8-oxoG was used to stimulate DCs, and cellular function was investigated. The results revealed that 22 genes associated with DC function were significantly upregulated in mouse lung tissue after a single stimulation, and this increased to 42 genes upon further stimulation. Exposure of primary cultured human monocyte-derived DCs (moDCs) to 8-oxoG resulted in significantly enhanced expression of the cell-surface molecules CD40, CD86, CD83, and HLA-DQ, and increased secretion of the pro-inflammatory factors IL-6, TNF, and IL-8, but no significant changes in the anti-inflammatory cytokine IL-10. The stimulatory effect of 8-oxoG on human moDCs was abolished after silencing OGG1 [42]. However, blocking TLR-mediated signaling with a MyD88-specific inhibitory peptide (Pepinh-MYD) did not affect the activation of 8-oxoG-treated moDCs, indicating that TLRs in endosomes are not involved in this activation. Thus, the authors speculated that 8-oxoG-triggered DC activation occurs via the activation of small molecule GTPases by OGG1. In support of this, small molecule GTPases, such as Rac1 and RhoG, have been found to regulate gene expression upon exposure to ultraviolet radiation, DNA damage, and ROS, respectively, mainly via initiation of the MAPK and PI3K cascade reactions to amplify the signal, which ultimately activates the NF-κB signaling pathway [24,43]. The NF-κB pathway regulates inflammation through the regulation of pro-inflammatory cytokines and chemokines, such as IL-6 [44].

### 3.3. OGG1 and Granulocytes

Granulocytes are a heterogeneous group of leukocytes, including neutrophils, eosinophils, and basophils [45]. Granulocytes play a role in inflammation, autoimmune diseases, and cancer. Granulocytes serve as key components of innate immune cells under oxidative stress stimulation, and their normal function is essential for the maintenance of immunological functions.

Neutrophils, the most abundant type of granulocytes in the human blood, are the body’s first line of defense against the invasion of foreign pathogens. They play a significant role in inflammatory responses, including chemotaxis, phagocytosis, degranulation, and the production of neutrophil extracellular traps (NETs) [46]. Many studies have shown that neutrophils play a powerful role in immune regulation, particularly in inflammatory responses [47]. 

Studies have also explored how the presence of OGG1 affects the recruitment and activation of neutrophils in the case of oxidative stress. One study developed a mouse model of aseptic pneumonia using LPS and TNF-α. In this model, neutrophil infiltration in the mouse lung decreased rapidly after administration of TH5487, a small molecule inhibitor of OGG1. Moreover, this inhibitory effect persisted after nine hours of TNF-α stimulation when TH5487 was administered non-prophylactically. This suggests that the effect was time-dependent [22,48]. These results confirm the impact of OGG1 on neutrophil recruitment during the inflammatory response.

Bacsi et al. also demonstrated a link between OGG1 and neutrophils. Allergic airway inflammation was induced by ragweed pollen extract (RWPE) in mice with siRNA silencing of *Ogg1* (Ogg1^D^). The amount of 8-oxoG in the alveolar epithelial cells (AECs) of the control Ogg1^P^ mice was significantly reduced when compared to Ogg1^D^ mice after three hours of RWPE stimulation. This indicates that the presence of OGG1 effectively repaired 8-oxoG. However, the timing of neutrophil recruitment in response to stimulation and the formation of ROS following neutrophil activation were consistent with a second-phase increase in 8-oxoG content that occurred between 8 and 36 h after RWPE stimulation. Further investigation of DNA single-strand breaks (SSBs) upon RWPE stimulation revealed a significant increase in the abundance of SSB in the AECs of Ogg1^P^ mice during neutrophil recruitment between 8 and 24 h after RWPE stimulation. Activated neutrophils were co-cultured with mouse embryo fibroblast (MEF) cells and the immortalized type 2 mouse lung epithelial cell line (MLE-12) in vitro, resulting in higher levels of SSBs in Ogg1^P^ MLE-12 and MEF cells than in Ogg1^D^ cells. The authors hypothesized that neutrophils in mice, and cells co-cultured with activated neutrophils, may create SSBs that are related to Ogg1 activity [26]. 

Studies have shown that type I IFN is a key cytokine in the pathogenesis of systemic lupus erythematosus (SLE) [49]. The current understanding is that ROS production can cause oxidative damage to mtDNA, which leads to the production of type I IFN. In SLE, low-density neutrophils are an important source of oxidized mtDNA, which is released from activated neutrophils upon NET production [50,51]. Another study investigated whether the presence or absence of OGG1 has an impact on the initiation and progression of IFN-driven SLE. Tumurkhuu et al. treated wild-type (WT) and *Ogg1^−/−^* mice with pristane to induce SLE and measured the levels of 8-oxoG in the peritoneal lavage fluid of mice. The authors found that *Ogg1^−/−^* mice had significantly higher levels of 8-oxoG than WT mice. After prolonged pristane treatment for 10 months, the expression of *Ogg1* in the treated WT mice was significantly decreased when compared to the untreated group, and the number of circulating neutrophils was significantly increased in treated WT and *Ogg1^−/−^* mice. Calcium ionophore A23187 can increase the concentration of intracellular calcium ions; this induces Ca^2+^-dependent cell death and can lead to the production of intra- and extracellular ROS. Hence, A23187 was used to stimulate bone marrow-derived neutrophils (BMDNs) from WT and *Ogg1^−/−^* mice treated with pristane. This resulted in increased formation of NETs over time. Moreover, the results revealed that NETs formed earlier and in a more pronounced manner in *Ogg1^−/−^* BMDN than in the WT BMDM. The above findings demonstrate that OGG1 loss aggravates oxidative damage in cells and, to some extent, affects the release of NETs upon neutrophil death to a certain extent [52].

As a type of granulocyte, eosinophils play an important role in inflammatory and allergic reactions and in the defense against parasitic, fungal, bacterial, and viral infections [53]. Eosinophils are considered central effector cells in asthma-related studies and are important clinical targets for the resolution of airway hyperresponsiveness [54]. 

Studies have also investigated how oxidative stress causes changes in eosinophil function and whether this involves OGG1. In order to explore the relationship between OGG1 and allergic inflammatory reactions, researchers exposed Ogg1^P^ mice to RWPE to induce allergic airway inflammation. The results revealed that eosinophil recruitment occurred in the lungs of mice after sensitization, while eosinophil recruitment was significantly reduced in the lungs of Ogg1^D^ mice. Direct stimulation of mouse lungs using 8-oxoG, which mimics the OGG1-BER pathway, was further investigated using RNA sequencing analysis. The results revealed the differential expression of 84 microRNAs (miRNA) in mouse lungs after sensitization. Exogenous administration of a down-regulated let-7b-p3 miRNA mimic or an upregulated inhibitor of miR-23a or miR-27a was able to reduce eosinophil recruitment and the production of mucus and collagen by controlling the expression of IL-4, IL-5, and IL-13. Therefore, it was concluded that the regulation of antigen-driven allergic immune responses by OGG1 signaling is mediated by the differential expression of Th2 cytokines and miRNAs upstream of eosinophils [55]. However, it is unclear how OGG1 regulates the function of neutrophils and eosinophils upon oxidative stress. Based on the important role of granulocytes in the occurrence and development of inflammation, the relationship between OGG1 and granulocytes requires further investigation.

### 3.4. OGG1 and Lymphocytes

Lymphocytes play an important role in acquired immunity. Studies have shown that ROS can inhibit the proliferation and function of T cells under high-oxygen culture conditions [56]. It has also been shown that excessive ROS can lead to the attenuation of T cell receptor (TCR)-induced Ca^2+^ influx, making lymphocytes unresponsive to mitogens and reducing their cell division capacity [57,58,59] while also upregulating the expression of the apoptosis-inducing molecules Fas and FasL, causing lymphocyte apoptosis. These results suggest that oxidative stress can affect T cell function.

Von der Lippen et al. found that human peripheral blood lymphocytes have a low efficiency in the repair of 8-oxoG accumulation caused by a photosensitizing stimulus (less than 10% clearance of 8-oxoG within 24 h). However, stimulation of lymphocytes with phytohemagglutinin (PHA) can activate T cell receptor signaling and significantly accelerate the repair ability. The expression of OGG1 was examined within 24 h of PHA stimulation prior to lymphocyte proliferation. The results demonstrated that the expression of OGG1 was upregulated nearly four-fold due to the increased binding of the transcription factor NF-YA and the OGG1 gene promoter. Jun kinase inhibitors can inhibit the binding of OGG1 to NF-YA and the subsequent induction of OGG1 [60], suggesting that human lymphocytes can repair oxidative damage by activating NF-YA to mediate the up-regulation of OGG1 and removal of 8-oxoG (Figure 2).

Gautam et al. isolated lymphocytes from blood samples of healthy volunteers across different age groups and evaluated the antioxidant status of lymphocytes by detecting lymphocyte lipids and protein damage. The results showed that oxidative damage to lymphocytes in elderly subjects was significantly increased when compared to young subjects [61]. This indicates that the balance of the oxidative/antioxidative system in lymphocytes is altered during aging, thereby accelerating oxidative damage. This further suggests that oxidative stress in lymphocytes may play an important role in immune senescence.

Similar findings have been reported in Alzheimer’s disease (AD), a disease closely related to aging. AD is a neurodegenerative disease accounting for more than 50% of all dementia cases [62]. A considerable body of evidence indicates that AD patients have a decreased DNA damage repair capacity. To investigate the effect of OGG1 gene polymorphisms on the DNA damage repair capacity of AD patients, OGG1 genotyping was performed, including an assessment of basal DNA damage, oxidative damage, and plasma 8-oxoG levels in lymphocytes of individuals with different genotypes. The results revealed that the level of DNA oxidative damage in the lymphocytes of OGG1 (Ser326Cys + Cys326Cys) gene carriers was higher than that of OGG1 (Ser326Ser) gene carriers [63]. When comparing the proportion of lymphocytes in the peripheral blood between AD patients and healthy volunteers, it was found that lymphocytes were significantly reduced in AD patients [64]. Dezor et al. detected the expression of OGG1, p53, and TNF-α in the peripheral blood lymphocytes of AD patients and healthy volunteers of the same age [65]. The results demonstrated that the levels of OGG1 and TNF-α proteins in the lymphocytes of AD patients were significantly decreased, as compared to the control group, while the protein levels of p53 were significantly increased. This suggests that the DNA repair proteins OGG1 and p53 and the inflammatory protein TNF-α may be involved in the pathogenesis of AD. Interestingly, in the early stages of neurodegenerative disease development, the presence of OGG1 promotes the downregulation of TNF-α levels. Moreover, while p53 expression is elevated during the early stages of AD, this expression and the associated DNA repair role gradually decrease with disease progression. However, the regulation of lymphocyte function by OGG1 and the underlying mechanisms during AD development remain to be explored.

Chronic inflammation is inextricably linked to lymphocyte development and function, and research indicates that chronic inflammation plays a critical role in tumorigenesis and metastasis through a variety of mechanisms [66]. A large body of evidence indicates that DNA damage caused by oxidative stress is strongly associated with the occurrence and progression of cancer [67]. Therefore, exploring the impact of OGG1 on the function of lymphocytes during tumorigenesis may lead to a new therapeutic approach.

Epidemiological studies have shown a close relationship between smoking and cancer. *Ogg1* and *Myh* double-knockout mice were used to investigate the role of DNA damage repair in cigarette smoke-induced cancers. The experimental results revealed that the number of DNA SSBs and the probability of chromosomal aberrations in the peripheral blood lymphocytes of *Ogg1^−/−^Myh^−/−^* mice were significantly increased after exposure to cigarette smoke extract (CSE) for 24 h, as compared to WT mice. Moreover, in vitro experiments revealed a significant increase in the DNA double-strand break marker H2AX in the peripheral blood lymphocytes after CSE stimulation for 3 h. These results suggest that lymphocyte function is impaired and immune surveillance is reduced after CSE stimulation, and this may be related to tumorigenesis. In addition, the survival rate of *Ogg1^−/−^Myh^−/−^* mice was reduced after exposure to the carcinogen benzo[a]pyrene, as compared to WT mice [68]. This result appears consistent with epidemiological data, indicating that a deficiency in DNA oxidative damage repair ability in lymphocytes increases the susceptibility to lung cancers [69].

Epstein–Barr virus (EBV) infection causes the immortalization of human B lymphocytes and is implicated in the pathogenesis of malignant tumors. EBV nuclear antigen (EBNA)-1 can induce the accumulation of ROS, leading to oxidative DNA damage. Researchers have found that the gene expressions of MTH1, OGG1, and MUTYH, which are related to oxidative damage repair, were upregulated in primary B cells after infection with EBV, and treatment with MTH1 inhibitors prevented B cell immortalization [70]. These results demonstrate that antioxidant treatment of cells can alleviate the negative effects of EBV infection. This provides new possibilities for the treatment of EBV-related cancers.

## 4. Conclusions

A large body of literature has shown that DNA oxidative damage is associated with many diseases. Accordingly, researchers have explored the repair of DNA oxidative damage, and the role of OGG1 in oxidative stress has been highlighted. Both endogenous and exogenous ROS can lead to impaired immune cell function, and as an integral part of the initiation of the BER pathway, numerous studies have demonstrated that OGG1 plays an important role in maintaining homeostasis of the body’s immune system. This undoubtedly demonstrates the great research potential and possible clinical value of OGG1. In particular, more recent findings have disrupted our traditional understanding of the functions of OGG1, suggesting that there are many potential functions of OGG1 that remain unexplored.

The current findings revealed that OGG1 plays a dual role in diseases such as inflammation. For example, in mouse models of atherosclerosis, systemic lupus erythematosus, and gastrointestinal inflammation, *Ogg1* deficiency exacerbates the inflammatory response and accelerates the disease process [28,71,72], whereas, in allergic airway inflammation and asthma, *Ogg1* acts as a pro-inflammatory agent [34,73]. Thus, it is clear that the role of OGG1 in different inflammatory settings or at different stages of the inflammatory response is not fully understood. The specific mechanisms by which OGG1 affects the function of immune cells and its particular role in maintaining the immune homeostasis of the organism are yet to be fully explored and elucidated by researchers.

## Figures and Tables

**Figure 1 antioxidants-12-01300-f001:**
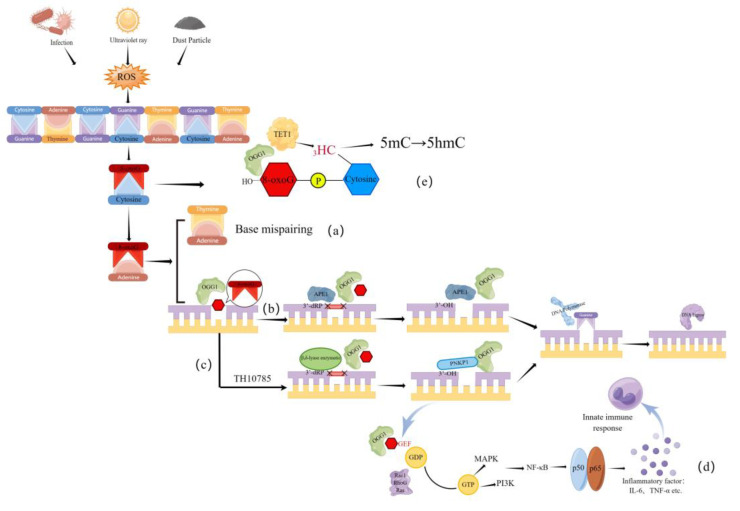
The function and mechanism of OGG1. (**a**) Oxidative stress can lead to base mismatch. (**b**) OGG1-BER pathway diagram. (**c**) TH10785 can activate the new lyase function of OGG1. (**d**) The function and mechanism of OGG1 in innate immunity. (**e**) OGG1 can act as a signaling molecule to regulate DNA demethylation. Abbreviations: APE1, AP nucleic acid endonuclease 1; PNKP1: polynucleotide kinase phosphatase; GEF: Guanine-Nucleotide Exchange Factor; GDP, Guanosine diphosphate; GTP, Guanosine triphosphate; MAPK, Mitogen-activated protein kinases; PI3K, Phosphoinositide 3-Kinase; NF-κB, nuclear factor kappa-B pathwaypathway; IL-6, Interleukin-6; TNF-α, Tumor Necrosis Factor alpha; TET1, Tet Methylcytosine Dioxygenase 1; 5mC, 5-methylcytosine; 5hmC, 5-hydroxymethylcytosine (Created with Figdraw).

**Figure 2 antioxidants-12-01300-f002:**
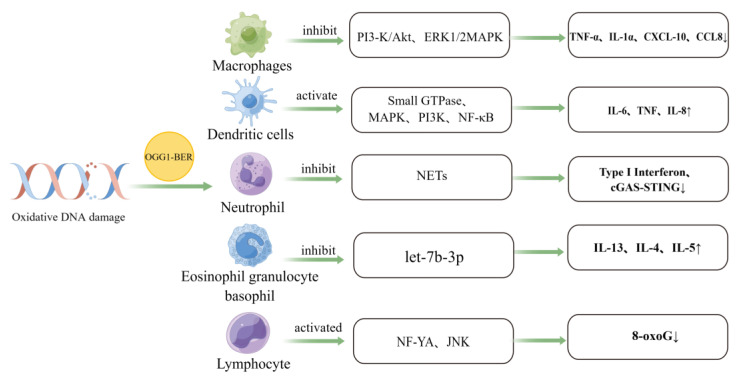
The effect of OGG1-BER pathway on immune cells. OGG1 can initiate the BER pathway to repair oxidatively damaged DNA, a process that affects the function of immune cells, including macrophages, dendritic cells, granulocytes, and lymphocytes. Immune cells are regulated by participating in a variety of signaling pathways (Created with Figdraw).

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
