# Peer review of "Effect of 8-Hydroxyguanine DNA Glycosylase 1 on the Function of Immune Cells"

_antioxidants, 2023, doi:10.3390/antiox12061300_

Round 1
Reviewer 1 Report
This paper provides insightful observations on the impact of 8-oxoguanine DNA glycosylase 1 (OGG1) on immune cell function. OGG1 is an essential enzyme involved in the repair of oxidative DNA damage, and its role has been extensively studied. The paper summarizes the importance of OGG1 in maintaining immune homeostasis. Additionally, it discusses the significance of antioxidants and the consequences of excessive reactive oxygen species (ROS) production within the body. The effects of oxidative stress on immune cell function and its involvement in aging and neurodegenerative diseases, such as Alzheimer's disease, are highlighted. It is worth noting that some readers may not be familiar with OGG1. Therefore, it would be beneficial to include a basic signaling pathway diagram illustrating the functions of OGG1 and the impact of oxidative stress, enabling readers to grasp the content more concretely.
Furthermore, in Figure 2, while the paper summarizes the immune cell types and their relationship to the OGG1-BER pathway, it would be helpful to provide a brief description of how the altered expression of these genes influences specific biological outcomes.
There are no issues with the English language.
Author Response
Dear Reviewer,
Thanks very much for taking your time to review this manuscript. We have uploaded the attachment,please see the attachment.
Thank you again!
Weiran Zhang

Reviewer 2 Report
Effect of 8-hydroxyguanine DNA glycosylase 1 on the function of immune cells is an interesting update on the roles of OGG1 in immune function. The English grammar is OK. However, some word choices made in the text left me a bit confused. This was not over the main thesis that OGG1 had effects on immune responses but on the directions of the impact in a few cases.
Figure 1 could be expanded to include the OGG1 + 8oxoG interaction with GTPases and OGG1 interaction with APE1 and TET1 (others?).
Lines 49-50; and the incidence of diabetes are all reduced in Ogg1-/- mice, ? My reading of reference 11 suggests the opposite. Also, is there a better reference than the one used as #12. Book chapters are often hard to access.
Lines 58-60; damage is crucial for maintaining functional cell homeostasis. Among the four bases, guanine (G) has the lowest oxidation potential and is the most easily oxidized base. The phrase, redox potential, or reduction potential would be better.
Lines 124-128 seem to repeat information mentioned in the paragraphs above.
Line 216-MLE-12, define as cell line and state origin.
Line 239-I was really confused by the comments on NETs. This line seems to contradict the above statement that NETS formed earlier and were more pronounced. Please clarify.
Line 347-How so? Do you mean, we have a better understanding of the threat of oxidative damage to human health?
Line 352-360-This is not a summary. It has new information which should have been mentioned above. Move the current comments elsewhere and rewrite the section.
Line 361- Oxidative DNA damage is now----
The actual grammar is well done. Something may have lost in translation relating to word choices outside of the actual grammar.
Author Response

(The authors gave the same response as above.)

Round 2
Reviewer 2 Report
The revised manuscript is acceptable for publication